# Prevalence and Correlates of Heavy Episodic Alcohol Consumption among Adults in Ecuador: Results of the First National STEPS Survey in 2018

**DOI:** 10.3390/ijerph17239017

**Published:** 2020-12-03

**Authors:** Supa Pengpid, Karl Peltzer

**Affiliations:** 1ASEAN Institute for Health Development, Mahidol University, Salaya 73170, Thailand; supaprom@yahoo.com; 2Department of Research Administration and Development, University of Limpopo, Turfloop 0727, South Africa; 3Department of Psychology, University of the Free State, Bloemfontein 9300, South Africa

**Keywords:** heavy episodic drinking, sociodemographic factors, health variables, Ecuador

## Abstract

Thise study aimed to assess the prevalence and correlates of heavy episodic drinking (HED) among adults in Ecuador. In the national, cross-sectional 2018 Ecuador STEPwise approach to Surveillance (STEPS) survey, 4638 persons (median age = 39 years, range 18–69 years) responded to a questionnaire and physical measures. Logistic regression was used to assess the determinants of HED. Results indicate that 24.1% had past-month HED, 36.7% among men, and 12.0% of women; among past-12-month drinkers, 40.6% had past-month HED. In adjusted logistic regression analysis, male sex (adjusted odds ratio = AOR: 3.03, 95% confidence interval = CI: 2.44–3.77), past smoking (AOR: 1.42, 95% CI: 1.12–1.81), and current smoking (AOR: 2.94, 95% CI: 2.25–3.86) were positively associated with HED, and being aged 50–69 years (AOR: 0.52, 95% CI: 0.39–0.68) was negatively associated with HED. In sex-stratified analyses among men, being African Ecuadorean or Mulato (AOR: 1.74, 95% CI: 1.07–2.84) and high physical activity (AOR: 1.43, 95% CI: 1.02–2.01) were positively associated with HED, and among women, being Montubia (AOR: 0.38, 95% CI: 0.16–0.90) was negatively associated with HED and obesity (AOR: 1.58, 95% CI: 1.05, 2.38) was positively associated with HED. Almost one in four participants engaged in HED, and several sociodemographic and health indicators were identified associated with HED.

## 1. Introduction

Harmful alcohol use is a significant contributor to the global burden of disease [1]. Heavy episodic drinking (HED) is defined as “drinking at least 60 g or more of pure alcohol on at least one occasion in the past 30 days,” and is a major indicator for the acute effects of alcohol consumption, such as injury [2]. Worldwide, 18.2% of persons 15 years and older were estimated to engage in HED in 2016 [3], and HED is expected to increase to 23% in 2030 [4]. In the Americas region, the prevalence of HED (≥15 years) decreased from 29.4% in 2000 to 21.3% in 2016 [3], and in 2016 in Columbia it was 15.4%, in Peru 26.3%, in Argentina 23.0%, in Bolivia 21.1%, in Brazil 19.7%, and in Ecuador 21.3% [3]. In national population-based surveys, the prevalence of HED was, among regular alcohol users, 46% in Brazil [5], and among the general population, 12.5% in Ethiopia [6] and 12.8% in Kenya [7]. In the World Health Survey in Ecuador in 2003, 5.5% were heavy drinkers (≥1 day, ≥5 standard drinks/past 7 days) [8]. In a study among university students in Ecuador, the prevalence of hazardous or harmful alcohol use was 23.8% among females and 49.7% in males [9]. There is a lack of national population-based data on the prevalence and correlates of HED in Ecuador.

Ecuador is an upper middle-income country and has a total population of 16.9 million; the largest ethnic group is Mestizo (people of mixed Indian and European descent) (71.9%), followed by Montubio (an aboriginal mestizo group that originates from the coastal part of Ecuador) (7.4%), Amerindian (or indigenous people) (7%), white (6.1%), and Afroecuadorian (4.3%), and the majority are Roman Catholic (74%) and Evangelical (10.4%) by religion [9]. The urban population is 64.2%, life expectancy at birth is 77.5 years, and 92.8% can read and write [9]. “Ecuador’s high poverty and income inequality most affect indigenous, mixed race, and rural populations” [10].

Factors associated with HED may include male sex [6,7,11,12,13,14,15], middle adulthood [7,16,17], specific ethnic groups [7,17,18], lower socioeconomic status [5,17,19,20,21], urban residence [17], rural residence [6], former and current tobacco use [6,7,8,12,17,22], overweight/obesity [23,24,25], sedentary behavior [5], physical activity [26,27,28], inadequate fruit and vegetable consumption [29], and less frequent primary healthcare utilization [30,31]. Epidemiological population-based surveys are needed to target interventions to prevent HED. The study aimed to assess the prevalence and correlates of HED among adults in a nationally representative survey in Ecuador in 2018.

## 2. Methods

### 2.1. Study Design and Participants

Cross-sectional nationally representative data of 18–69 year-old persons from the 2018 Ecuador STEPS Survey were analyzed [32]. A three-stage cluster sampling strategy was used. First stage: selection of primary sampling units (PSU) by stratum; second stage: selection of 12 occupied households within each PSU selected in the first stage; and the third stage: random selection of 1 participant aged 18–69 years from each household [33]. The overall study response rate was 69.4% [33]. The data were collected using electronic tablet devices [33]. The Ethical Review Committee of the Ecuador Ministry of Health provided ethical approval of the study, and written informed consent was obtained from the study participants [33].

### 2.2. Measures

#### Alcohol Use Variables

A standard unit of alcoholic beverage was defined as 10 g of pure ethanol (using showcards and examples), and questions included [33]: “Have you ever consumed any alcohol such as beer, wine, spirits, etc.?” (yes, no); “Have you consumed any alcohol within the past 12 months?” (yes, no); “Have you stopped drinking due to health reasons, such as a negative impact on your health or on the advice of your doctor or other health workers?” (yes, no); “Have you consumed any alcohol within the past 30 days?” (yes, no); “During the past 30 days, how many times did you have six or more standard drinks in a single drinking occasion?” (number of times); past-week alcohol consumption was measured with seven questions on the drinks consumed on each day of the week (number); “During the past 7 days, did you consume any homebrewed alcohol, any alcohol brought over the border/from another country, any alcohol not intended for drinking or other untaxed alcohol?” (yes, no); “During the past 12 months, have you had family problems or problems with your partner due to someone else’s drinking?” (1 = yes: > monthly to 4 = once or twice). Alcohol dependence was assessed with three questions of the Alcohol Use Disorder Identification Test, AUDIT (items 4–6); e.g., “How often during the last year have you found that you were not able to stop drinking once you had started?” Response options ranged from “0 = never to 4 = daily or almost daily”; total scores of 4 or more indicate alcohol dependence [34].

Socio-demographic factor questions included age (in years), sex (male, female), highest level of education completed (1 = no formal school to 7 = postgraduate degree), and ethnicity (Mestizo, Amerindian, Montubio, African-Ecuadorian, whites, etc.) [33].

The health risk behavior variables included past and current smoking, consumption of fruit and vegetables per day, and “based on the “Global Physical Activity Questionnaire,” sedentary behavior (≥7 h/day), and low, moderate, or high physical activity [33]. Body mass index was measured: “<18.5 kg/m^2^ underweight, 18.5–24.4 kg/m^2^ normal weight, 25–29.9 kg/m^2^ overweight, and ≥30 kg/m^2^ obesity” [33]. One question was asked about a past-12-month healthcare provider visit [33].

### 2.3. Data Analysis

All statistical analyses were conducted using STATA software version 14.0 (Stata Corporation, College Station, TX, USA). The data were weighted to make the sample representative of the target population in Ecuador. Descriptive statistics were used to summarize the sample and HED prevalence characteristics. Unadjusted and adjusted (including variables significant at *p* < 0.1 in univariate analysis) logistic regression was used to predict HED prevalence. In addition, sex-stratified multivariable logistic regression models for HED prevalence were calculated. Based on a previous literature review [5,6,7,8,12,16,17,19,20,21,22,23,24,25,26,27,28,29,30,31], covariates included sex, age group, highest level of education, ethnicity, smoking status, obesity, fruit and vegetable intake, physical activity, sedentary behavior, and healthcare utilization. Taylor linearization methods were applied to account for the complex study design and the sampling weight. Results from logistic regression analyses are reported as odds ratios (ORs) and 95% confidence intervals (Cis). Missing values (<3.6% for any study variable) were excluded and *p* < 0.05 considered significant.

## 3. Results

### 3.1. Characteristics of the Sample and Heavy Episodic Alcohol Consumption

The sample comprised 4638 persons 18 to 69 years (median = 39 years, IQR = 27–52); 1944 (41.9%) were men and 2694 (58.1%) were women; 51.7% had secondary or higher education; and 78.9% were Mestizo by ethnicity. Almost one in four were past (24.4%) or current smokers (13.7%), 58.8% had 0–1 servings of fruit and vegetables per day, 49.8% had high physical activity, 12.4% had sedentary behavior, 25.7% were obese, and 59.4% had visited a healthcare provider in the past 12 months. Almost one in four participants (24.1%) had HED, 36.7% among men and 12.0% of women (see Table 1).

### 3.2. Distribution of Alcohol Use Pattern

More than four in five participants (83.2%) had ever consumed alcohol, 59.8% had in the past 12 months, and 39.3% had in the past month. Infrequent (1–2 days) HED was 18.0%, and frequent (≥3 days) HED was 6.9%. Among past-12-month alcohol users, HED was 40.6%, and heavy alcohol use in the past 7 days (1 or more days ≥5 standard drinks/day) was 11.7%. The prevalence of alcohol dependence was 11.9%; 7.0% had family alcohol problems in the past year, 8.8% stopped drinking because of health reasons, and concurrent past-month HED and past-month smoking was 6.9% (see Table 2).

### 3.3. Associations with HED

In adjusted logistic regression analysis, male sex (adjusted odds ratio = AOR: 3.03, 95% confidence interval = CI: 2.44–3.77), past smoking (AOR: 1.42, 95% CI: 1.12–1.81), and current smoking (AOR: 2.94, 95% CI: 2.25–3.86) were positively associated with HED, and being aged 50–69 years (AOR: 0.52, 95% CI: 0.39–0.68) was negatively associated with HED. Furthermore, in the unadjusted analysis, healthcare provider visit in the past 12 months was negatively associated with HED (see Table 3). In addition, in sex-stratified analyses among men, being African Ecuadorean or Mulato (AOR: 1.74, 95% CI: 1.07–2.84) and high physical activity (AOR: 1.43, 95% CI: 1.02–2.01) were positively associated with HED, and among women being Montubia (AOR: 0.38, 95% CI: 0.16–0.90) was negatively associated with HED and obesity (AOR: 1.58, 95% CI: 1.05, 2.38) was positively associated with HED (see Table 4).

## 4. Discussion

For the first time in this national community-based survey among adults 18–69 years old in Ecuador, a high prevalence of past-month HED among the general population (24.1%) was found. The HED prevalence we found in Ecuador was higher than estimates in the Americas region (21.3%) in 2016 [3], Columbia (15.4%) [3], Argentina (23.0%) [3], Bolivia (21.1%) [3], Brazil (19.7%) [3], Ecuador (21.3%) [3], Ethiopia (12.5%) [6], and Kenya (12.8%) [7], and lower than among regular alcohol users in Brazil (46%) [5]. The prevalence of past-month alcohol use (39.3%) in this study was similar to a national sample of 20–59 year-olds in Ecuador in 2012 (41.3%) [35]. Compared to the 2003 Ecuador World Health Survey (5.5% past-week heavy drinkers) [8], this study in 2018 showed a prevalence of 11.7% past-week heavy drinkers. The possible increase of alcohol consumption in Ecuador is also shown in the high prevalence of hazardous or harmful alcohol use (23.8% among females and 49.7% in males) among university students in Ecuador [9]. The overall increase of heavy drinking in Ecuador from 2003 to 2018 may be related to a larger middle class and economic development in Ecuador, as “alcohol consumption and resulting problems are likely to rise with increasing income” [36]. The World Health Organization [36] (p. 2) notes that, “traditional drinking patterns dominated by sporadic episodes of intoxication continue, but involvement in the cash economy and industrialization of alcohol production and distribution have permitted the episodes to become more frequent.”

In agreement with previous studies [6,7,11,12,13,14,15,16,17], this study found that male sex increased the odds and older age (50–69 years) decreased the odds of engaging in HED. Sex specific role expectations and norms, such as associating drinking alcohol with masculinity, may be related to the male preponderance of HED [6,14]. At an older age, a reduction of HED may be expected since the tolerance towards alcohol reduces with ageing [37]. Among different ethnic groups in Ecuador, African Ecuadorean or Mulato men had more HED, and Montubia women had less HED. It is possible that the legacy of slavery and marginalization of African Ecuadorean or Mulato (mixed between African and white) men in Ecuador increases their risk of HED [38], and since Montubia women mostly inhabit the rural areas of the coastal provinces in Ecuador, they have a lower risk for HED [39]. While previous research showed associations involving lower education or lower socioeconomic status [5,17,19,20,21], this study did not find that educational level was associated with HED.

Consistent with previous research findings [6,7,8,12,17,22], this study found strong associations between past and current smoking and HED. Reasons for this may lie in the co-dependence risk of HED and nicotine [40]. Public health interventions should be directed at integrating smoking cessation in persons with HED. A small proportion of participants in this study (8.8%) had stopped drinking due to health reasons. Therefore, screening and brief intervention of alcohol problems in primary care should be reinforced.

Having visited a healthcare facility in the past 12 months was in an unadjusted analysis protective against HED, which is consistent with previous research [30,31]. In a study in Brazil, people who were highly health conscious were less likely to engage in HED [5]. Some studies [5] showed a clustering of HED with other health risk behaviors, apart from tobacco use, such as sedentary behavior [5], physical activity [26,27,28], inadequate fruit and vegetable consumption [29], and obesity [23,24,25], while in this study physical activity among men and obesity among women were associated with HED, but no associations were found for inadequate fruit and vegetable intake, or sedentary behavior. Interventions targeting obesity in women in Ecuador should include HED, as HED is related to caloric content [23].

In line with the World Health Organization’s [41] recommendations, Ecuador has incorporated in its Strategy for the Prevention and Control of Harmful Alcohol Consumption, three measures: reduce the ability to purchase due to less availability—for example, reduce hours and sales in public places; reduce exposure to advertising; and increase taxes and prices. However, challenges remain to effectively implement policies for the prevention and control of harmful consumption of alcohol, such as reducing hours and spending in public places, reducing advertising, and increase taxes and prices [33].

### Study Limitations

The study was limited by its cross-sectional design and the self-reporting of some data, including alcohol use. Some variables, such as household income, had too much missing data and could therefore not be included in the analysis. A further limitation was that in this household survey, heavy drinking populations, such as military personnel, the homeless, and institutionalized persons, were not included [42].

## 5. Conclusions

Almost one in four participants engaged in HED; two in five among current drinkers engaged in HED; and sociodemographic factors (male sex, 18–29 and 30–49 year-olds and ethnicity) and health factors (smoking, body weight status, and physical activity) were associated with HED.

## Figures and Tables

**Table 1 ijerph-17-09017-t001:** Characteristics of the sample and heavy episodic drinking (HED) among adults in Ecuador, 2018.

Variable (#Missing Cases)	Sample	HED		
		Male	Female	Both sexes
	N (%) ^a^	% ^a^	% ^a^	% ^a^
All	4638	36.7	12.0	24.1
Age in years (#0)				28.6
18–29	1205 (30.1)	39.6	17.4	17.4
30–49	2034 (40.7)	41.1	11.6	25.2
50–69	1399 (29.3)	28.4	7.0	18.0
Education (#4)				
<Secondary	2452 (48.2)	37.8	9.2	22.8
Secondary	929 (21.2)	34.5	13.3	24.1
>Secondary	1253 (30.5)	36.6	15.9	26.3
Ethnicity (#2)				
Mestizo	3567 (78.9)	36.7	12.8	24.2
Amerindian	378 (6.5)	26.9	9.3	17.7
Montubio	335 (7.2)	38.1	4.6	24.0
African-Ecuadorian or Mulato	227 (4.4)	50.6	11.8	31.0
Whites or other	129 (3.0)	35.1	13.4	24.9
Smoking status (#0)				
Never	3046 (61.9)	30.4	10.2	16.7
Past	1007 (24.4)	32.7	17.1	28.0
Current	585 (13.7)	53.6	32.3	50.4
Fruit/vegetables (servings/day) (#11)				
0–1	2754 (58.8)	37.2	12.1	24.7
2–3	1416 (31.2)	34.4	11.3	21.9
4 or more	457 (10.0)	40.6	13.6	27.6
Physical activity (#6)				
Low	1083 (24.7)	30.9	13.3	20.0
Moderate	1206 (25.5)	30.7	11.5	18.8
High	2343 (49.8)	40.4	11.5	28.8
Sedentary behavior (#3)	481 (12.4)	37.7	14.6	27.7
Obesity (#168)	1204 (25.7)	39.6	13.0	23.5
Healthcare provider visit in past 12 months (#0)	2753 (59.4)	34.3	11.9	21.7

^a^ Percentage is calculated as the weighted percentage of complete cases.

**Table 2 ijerph-17-09017-t002:** Distribution of alcohol use pattern among adults in Ecuador, 2018.

Variable	Male	Female	Both Sexes
	*n* (%)	*n* (%)	*n* (%)
Ever alcohol use	1740 (90.4)	2050 (76.2)	3790 (83.2)
Alcohol use in the past 12 months	1401 (72.6)	1240 (47.6)	2641 (59.8)
Alcohol use in the past 30 days	973 (51.8)	674 (27.3)	1647 (39.3)
Heavy episodic drinking in the past 30 days			
0 days	1242 (62.5)	2373 (87.0)	3615 (75.0)
1 day	371 (19.3)	214 (8.6)	585 (13.8)
2 days	125 (7.1)	39 (1.5)	164 (4.2)
3 or more days	206 (11.1)	68 (3.0)	274 (6.9)
Alcohol dependence	396 (20.6)	97 (3.4)	493 (11.8)
Heavy alcohol use in the past 7 days (1 or more days ≥5 standard drinks)	480 (11.7)	355 (18.6)	125 (5.0)
Alcohol family problems in the past 12 months	153 (7.6)	159 (6.4)	312 (7.0)
Stopped drinking alcohol due to health reasons	168 (8.9)	253 (8.6)	421 (8.8)
Undeclared alcohol use	98 (5.0)	33 (1.0)	131 (3.0)
Past month HED and past-month smoking	246 (12.7)	27 (1.3)	273 (6.9)
HED among past-12-month alcohol users	682 (50.8)	297 (25.6)	979 (40.6)

**Table 3 ijerph-17-09017-t003:** Associations with heavy episodic alcohol consumption.

Variable	Simple Logistic Regression		Multiple Logistic Regression	
	Crude OR (95% CI)	*p*-Value	Adjusted OR (95% CI)	*p*-Value
Sex				
Female	1 (Reference)		1 (Reference)	
Male	4.25 (3.64, 5.10)	<0.001	3.03 (2.44, 3.77)	<0.001
Age in years				
18–29	1 (Reference)		1 (Reference)	
30–49	0.84 (0.69, 1.02)	0.082	0.87 (0.69, 1.10	0.253
50–69	0.55 (0.43, 0.70)	<0.001	0.52 (0.39, 0.68)	<0.001
Education				
<Secondary	1 (Reference)		1 (Reference)	
Secondary	1.08 (0.86, 1.35)	0.524	0.95 (0.75, 1.22)	0.696
>Secondary	1.21 (1.00, 1.47)	0.052	1.09 (0.87, 1.37)	0.462
Ethnicity				
Mestizo	1 (Reference)		1 (Reference)	
Amerindian	0.67 (0.40, 1.14)	0.139	0.64 (0.37, 1.10)	0.104
Montubio	0.98 (0.72, 1.34)	0.920	0.95 (0.69, 1.31)	0.750
African-Ecuadorian or Mulato	1.40 (0.97, 2.03)	0.071	1.34 (0.93, 1.95)	0.118
Whites or other	1.03 (0.63, 1.70)	0.894	1.05 (0.60, 1.85)	0.864
Smoking status				
Never	1 (Reference)		1 (Reference)	
Past	1.94 (1.56, 2.41)	<0.001	1.42 (1.12, 1.81)	0.008
Current	5.06 (4.01, 6.39)	<0.001	2.94 (2.25, 3.86)	<0.001
Fruit/vegetables (servings/day)				
0–1	1 (Reference)		1 (Reference)	
2–3	0.86 (0.71, 1.03)	0.095	0.91 (0.75, 1.12)	0.382
4 or more	1.17 (0.88, 1.55)	0.286	1.11 (0.83, 1.48)	0.468
Physical activity				
Low	1 (Reference)		1 (Reference)	
Moderate	0.93 (0.71, 1.20)	0.567	0.85 (0.65, 1.12)	0.255
High	1.62 (1.30, 2.03)	<0.001	1.18 (0.93, 1.50)	0.177
Sedentary behavior	1.24 (0.96, 1.60)	0.096	0.99 (0.75, 1.32)	0.946
Obesity	0.86 (0.68, 1.09)	0.858	-	
Healthcare provider visit in past 12 months	0.73 (0.61, 0.87)	<0.001	0.92 (0.75, 1.11)	0.371

OR = odds ratio; CI = confidence interval.

**Table 4 ijerph-17-09017-t004:** Multivariable logistic regression with heavy episodic alcohol consumption stratified by sex.

Variable	Men		Women	
	Adjusted OR (95% CI)	*p*-Value	Adjusted OR (95% CI)	*p*-Value
Age in years				
18–29	1 (Reference)		1 (Reference)	
30–49	1.07 (0.80, 1.42)	0.653	0.63 (0.44, 0.90)	0.012
50–69	0.64 (0.46, 0.89)	0.009	0.34 (0.20, 0.56)	<0.001
Education				
<Secondary	1 (Reference)		1 (Reference)	
Secondary	0.84 (0.61, 1.15)	0.273	1.21 (0.78, 1.90)	0.392
>Secondary	0.96 (0.70, 1.30)	0.778	1.34 (0.94, 1.92)	0.104
Ethnicity				
Mestizo	1 (Reference)		1 (Reference)	
Amerindian	0.60 (0.29, 1.21)	0.154	0.84 (0.49, 1.45)	0.527
Montubio	1.14 (0.79, 1.65)	0.483	0.38 (0.16, 0.90)	0.028
African-Ecuadorian or Mulato	1.74 (1.07, 2.84)	0.026	0.77 (0.38, 1.58)	0.483
Whites or other	1.06 (0.51, 2.19)	0.884	1.11 (0.50, 2.49)	0.794
Smoking status				
Never	1 (Reference)		1 (Reference)	
Past	1.21 (0.90, 1.62)	0.211	1.85 (1.24, 2.75)	0.003
Current	2.57 (1.94, 3.40)	<0.001	4.06 (2.35, 7.03)	<0.001
Fruit/vegetables (servings/day)				
0–1	1 (Reference)			
2–3			1 (Reference)	0.572
4 or more	1.16 (0.80, 1.69)	0.561	0.91 (0.65, 1.26)	0.903
Physical activity				
Low	1 (Reference)		1 (Reference)	
Moderate	0.95 (0.64, 1.43)	0.819	0.76 (0.52, 1.13)	0.180
High	1.43 (1.02, 2.01)	0.039	0.89 (0.60, 1.31)	0.552
Sedentary behavior	1.04 (0.72, 1.49)	0.837	0.92 (0.59, 1.34)	0.705
Obesity	1.28 (0.92, 1.77)	0.140	1.26 (0.92, 1.74)	0.154
Healthcare provider visit in past 12 months	0.88 (0.69, 1.13)	0.320	0.97 (0.71, 1.33)	0.850

OR = odds ratio; CI = confidence interval.

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
