# Peer review of "Prevalence and Correlates of Heavy Episodic Alcohol Consumption among Adults in Ecuador: Results of the First National STEPS Survey in 2018"

_ijerph, 2020, doi:10.3390/ijerph17239017_

Round 1

Reviewer 1 Report

Thank you for addressing the comments. I do not have any further comments. 

Author Response

OK

Reviewer 2 Report

This manuscript needs to be proof read for occasional errors in the text. One example is on p. 7 "Some variables in the Ecuador 2018 STEPS survey had too missing values, ..."

Author Response

Reviewer II
This manuscript needs to be proof read for occasional errors in the text. One example is on p. 7 "Some variables in the Ecuador 2018 STEPS survey had too missing values, ..."
Response: corrected

Reviewer 3 Report

Thank you for the opportunity to contribute to the pre-publication peer review process for the manuscript titled 'Prevalence and correlates of heavy episodic alcohol consumption among adults in Ecuador: Results of the first national STEPS survey in 2018' (ijerph-1015664).

The following relatively minor suggestions only are offered to assist improving the readability of the material for the IJERPH audience:

line 30 - suggest amending such that sentence does not start with an abbreviation.

lines 45-6 - suggest review font size for consistency.

line 68 - suggest review syntax 'ethanol or to pure alcohol' - not clear / missing a word?

Table 1 - suggest review highlighting - will this be a desired feature of final version (& within editorial style preference)?

line 120 - suggest amending such that sentence does not start with an abbreviation.

lines 144-8 - suggest amending this long and complex sentence into a couple of smaller distinct statements - firstly what was found, then secondly how it compares to existing data.

References - need to review formatting of this list - variable styles evident (fonts, line spacing, indenting, journal title abbreviating, capitalization of article title

Author Response

Reviewer III
Thank you for the opportunity to contribute to the pre-publication peer review process for the manuscript titled 'Prevalence and correlates of heavy episodic alcohol consumption among adults in Ecuador: Results of the first national STEPS survey in 2018' (ijerph-1015664).
The following relatively minor suggestions only are offered to assist improving the readability of the material for the IJERPH audience:
line 30 - suggest amending such that sentence does not start with an abbreviation.
Response: corrected
lines 45-6 - suggest review font size for consistency.
Response: corrected
line 68 - suggest review syntax 'ethanol or to pure alcohol' - not clear / missing a word?
Response: corrected
Table 1 - suggest review highlighting - will this be a desired feature of final version (& within editorial style preference)?
Response: corrected
line 120 - suggest amending such that sentence does not start with an abbreviation.
Response: corrected
lines 144-8 - suggest amending this long and complex sentence into a couple of smaller distinct statements - firstly what was found, then secondly how it compares to existing data.
Response: corrected
References - need to review formatting of this list - variable styles evident (fonts, line spacing, indenting, journal title abbreviating, capitalization of article title
Response: corrected

This manuscript is a resubmission of an earlier submission. The following is a list of the peer review reports and author responses from that submission.

Round 1

Reviewer 1 Report

Dear Authors,

The manuscript received for review raises a very important problem in the context of public health, i.e. prevalence and correlates of heavy episodic alcohol consumption among adults in Ecuador. The manuscript is interesting, although in my opinion it contains a few mistakes:

- the introduction covers the issue exhaustively, although it seems to me that the phrase "cardio-metabolic risk" is inconsistent with the applied medical nomenclature; metabolic syndrome, cardiovascular risk - yes

- there is a reference to the questionnaire in the abstract; in the description of the method no, there are only questions and variants of answers; this tool must be described in the method

- unfortunately, I don't understand the meaning of "(using showcards and examples)"

- I am also not sure whether alcohol dependence can be assessed selectively using questions from the AUDIT scale, i.e. whether the obtained results can be correctly interpreted on the basis of 3 out of 10 scale questions

- the results concerning associations with HED can be described as statistically significant or insignificant relationships

- there is no presentation of the results of the analysis "gender stratification vs HED" in the manuscript text, although they are discussed

- at the beginning of the discussion, the statistics from the introduction are unnecessarily duplicated, which I propose to delete

- in my opinion, the first line of conclusions are the results

- both in the conclusions and in the results, there are statements for age such as young age, middle age or older age. In tables 1 and 3, there are only age ranges that can only fit into these categories

- however, the data in Tables 1 and 2 raise the most doubts - why the sum of the numerical values ​​for individual variables does not equal the given number N = 4638, although% adds up to 100 or not, which is also a mistake; maybe a number N should be given for variables (?); what values ​​are excluded?; are the obtained results reliable?

- the literature requires updating, especially with regard to the oldest items

Best Regards

Author Response

Reviewer I

The manuscript received for review raises a very important problem in the context of public health, i.e. prevalence and correlates of heavy episodic alcohol consumption among adults in Ecuador. The manuscript is interesting, although in my opinion it contains a few mistakes:
- the introduction covers the issue exhaustively, although it seems to me that the phrase "cardio-metabolic risk" is inconsistent with the applied medical nomenclature; metabolic syndrome, cardiovascular risk – yes
Response: corrected, as below
Factors associated with HED may include male sex [6,7,11-15], middle adulthood [7,16,17], specific ethnic groups [7,17,18], lower socioeconomic status [5,17,19-21], urban residence [17], rural residence [6], former and current tobacco use [6-8,12,17,22], overweight/obesity [23-25], sedentary behaviour [5], physical activity [26-28], inadequate fruit and vegetable consumption [29], and less frequent primary health care utilization [30,31].
- there is a reference to the questionnaire in the abstract; in the description of the method no, there are only questions and variants of answers; this tool must be described in the method
Response: more is added
- unfortunately, I don't understand the meaning of "(using showcards and examples)"
Response: pictures of units of different types of alcoholic beverages are presented
- I am also not sure whether alcohol dependence can be assessed selectively using questions from the AUDIT scale, i.e. whether the obtained results can be correctly interpreted on the basis of 3 out of 10 scale questions
Response: According to the provided reference, yes
- the results concerning associations with HED can be described as statistically significant or insignificant relationships
Response: both are described
- there is no presentation of the results of the analysis "gender stratification vs HED" in the manuscript text, although they are discussed
Response: added, as below
Table 4. Multivariable logistic regression with heavy episodic alcohol consumption stratified by sex.
Variable Men Women
Adjusted OR (95% CI) p-value Adjusted OR (95% CI) p-value
Age in years
18-29
30-49
50-69
1 (Reference)
1.07 (0.80, 1.42)
0.64 (0.46, 0.89)

0.653
0.009
1 (Reference)
0.63 (0.44, 0.90)
0.34 (0.20, 0.56)

0.012
<0.001
Education
<Secondary
Secondary
>Secondary
1 (Reference)
0.84 (0.61, 1.15)
0.96 (0.70, 1.30)

0.273
0.778
1 (Reference)
1.21 (0.78, 1.90)
1.34 (0.94, 1.92)

0.392
0.104
Ethnicity
Mestizo
Amerindian
Montubio
African-Ecuadorian or Mulato
Whites or other
1 (Reference)
0.60 (0.29, 1.21)
1.14 (0.79, 1.65)
1.74 (1.07, 2.84)
1.06 (0.51, 2.19)

0.154
0.483
0.026
0.884
1 (Reference)
0.84 (0.49, 1.45)
0.38 (0.16, 0.90)
0.77 (0.38, 1.58)
1.11 (0.50, 2.49)

0.527
0.028
0.483
0.794
Smoking status
Never
Past
Current
1 (Reference)
1.21 (0.90, 1.62)
2.57 (1.94, 3.40)

0.211
<0.001
1 (Reference)
1.85 (1.24, 2.75)
4.06 (2.35, 7.03)

0.003
<0.001
Fruit/vegetables (servings/day)
0-1
2-3
4 or more

1 (Reference)
0.93 (0.72, 1.20)
1.16 (0.80, 1.69)

0.561
0.435

1 (Reference)
0.91 (0.65, 1.26)
0.97 (0.61, 1.55)

0.572
0.903
Physical activity
Low
Moderate
High
1 (Reference)
0.95 (0.64, 1.43)
1.43 (1.02, 2.01)

0.819
0.039
1 (Reference)
0.76 (0.52, 1.13)
0.89 (0.60, 1.31)

0.180
0.552
Sedentary behaviour 1.04 (0.72, 1.49) 0.837 0.92 (0.59, 1.34) 0.705
Obesity 1.28 (0.92, 1.77) 0.140 1.26 (0.92, 1.74) 0.154
Health care provider visit in past 12 months 0.88 (0.69, 1.13) 0.320 0.97 (0.71, 1.33) 0.850
OR=Odds Ratio; CI=Confidence Interval.

- at the beginning of the discussion, the statistics from the introduction are unnecessarily duplicated, which I propose to delete
Response: modified
- in my opinion, the first line of conclusions are the results
Response: modified
- both in the conclusions and in the results, there are statements for age such as young age, middle age or older age. In tables 1 and 3, there are only age ranges that can only fit into these categories
Response: corrected
- however, the data in Tables 1 and 2 raise the most doubts - why the sum of the numerical values for individual variables does not equal the given number N = 4638, although% adds up to 100 or not, which is also a mistake; maybe a number N should be given for variables (?); what values are excluded?; are the obtained results reliable?
Response: Missing cases are added, as in bloew

Variable (#missing cases)

All
Age in years (#0)
18-29
30-49
50-69
Education (#4)
<Secondary
Secondary
>Secondary
Ethnicity (#2)
Mestizo
Amerindian
Montubio
African-Ecuadorian or Mulato
Whites or other
Smoking status (#0)
Never
Past
Current
Fruit/vegetables (servings/day) (#11)
0-1
2-3
4 or more
Physical activity (#6)
Low
Moderate
High
Sedentary behaviour (#3)
Obesity (#168)
Health care provider visit in past 12 months (#0)

- the literature requires updating, especially with regard to the oldest items
Response: Some 15 updated references are added

Reviewer 2 Report

Prevalence and correlates of heavy episodic alcohol consumption among adults in Ecuador: Results of the first national STEPS survey in 2018

This paper aims to assess the prevalence and correlates of heavy episodic drinking (HED) among adults using a national cross-sectional 2018 Ecuador STEPS survey. The study identified several socio-demographic and health indicators correlated with heavy episodic drinking. The analysis was stratified by gender. It is not clear how current analysis contributes to the literature or aids policy.

  1. The conclusion of the study states that the findings of the analysis may facilitate public health interventions for reducing HED. However, policy interventions can only be drawn from causal analysis, and the cross-sectional nature of this study does not allow causal inference. For example, the study finds that underweight was negatively associated with HED. However, according to previous study cited by the authors, weight status is not the determinant of drinking, but on the contrary, drinking determines weight status. Therefore, if underweight is included as an explanatory variable in the logistic model, it is endogenous (it is correlated with the error term of the regression) and therefore it is not clear what policy should target – drinking adults or underweight adults – since the direction of causality is not clear. Same argument applies to other included health variables. Moreover, if endogenous regressors are included in the same model, the estimates of the direct/causal effect of other explanatory variables is biased and no policy conclusions can be drawn from such analysis.
  2. No clear rationale is provided for the inclusion of explanatory variables in the multiple logistic regression model. For example, it is not clear how adding salt in food or having diabetes is related to HED. Only one study, which used Kenyan data, is used a s a reference for the inclusion of several explanatory variables. It is not clear how that study is applicable in Ecuadorian context. Need to have a comprehensive review of literature to motivate inclusion of covariates of HED.

Author Response

This paper aims to assess the prevalence and correlates of heavy episodic drinking (HED) among adults using a national cross-sectional 2018 Ecuador STEPS survey. The study identified several socio-demographic and health indicators correlated with heavy episodic drinking. The analysis was stratified by gender. It is not clear how current analysis contributes to the literature or aids policy.
1. The conclusion of the study states that the findings of the analysis may facilitate public health interventions for reducing HED. However, policy interventions can only be drawn from causal analysis, and the cross-sectional nature of this study does not allow causal inference.
Response: This is modified to below
Almost one in four participants engaged in HED, and several sociodemographic and health indicators were identified associated with HED.

2. For example, the study finds that underweight was negatively associated with HED. However, according to previous study cited by the authors, weight status is not the determinant of drinking, but on the contrary, drinking determines weight status. Therefore, if underweight is included as an explanatory variable in the logistic model, it is endogenous (it is correlated with the error term of the regression) and therefore it is not clear what policy should target – drinking adults or underweight adults – since the direction of causality is not clear. Same argument applies to other included health variables. Moreover, if endogenous regressors are included in the same model, the estimates of the direct/causal effect of other explanatory variables is biased and no policy conclusions can be drawn from such analysis.
3. No clear rationale is provided for the inclusion of explanatory variables in the multiple logistic regression model. For example, it is not clear how adding salt in food or having diabetes is related to HED. Only one study, which used Kenyan data, is used a s a reference for the inclusion of several explanatory variables. It is not clear how that study is applicable in Ecuadorian context. Need to have a comprehensive review of literature to motivate inclusion of covariates of HED.
Response to 2 and 3: some covariates are removed, and comprehensive review of the covariates added, as below.
Factors associated with HED may include male sex [6,7,11-15], middle adulthood [7,16,17], specific ethnic groups [7,17,18], lower socioeconomic status [5,17,19-21], urban residence [17], rural residence [6], former and current tobacco use [6-8,12,17,22], overweight/obesity [23-25], sedentary behaviour [5], physical activity [26-28], inadequate fruit and vegetable consumption [29], and less frequent primary health care utilization [30,31].

Reviewer 3 Report

This is an interesting study and important to investigate. The method section was insufficient,  in particular, the data analysis. Can you please rewrite this section to reflect the overall statistical analysis? In addition, the adjusted analysis was presented but the author did not mention the variables adjusted in the data analysis section. Please explain the method section clearly in the abstract too. 

Author Response

This is an interesting study and important to investigate. The method section was insufficient, in particular, the data analysis. Can you please rewrite this section to reflect the overall statistical analysis? In addition, the adjusted analysis was presented but the author did not mention the variables adjusted in the data analysis section. Please explain the method section clearly in the abstract too.
Response: below is added in the data analysis section, also additions in the abstract
All statistical analyses were conducted using STATA software version 14.0 (Stata Corporation, College Station, TX, USA). The data were weighted to make the sample representative of the target population in Ecuador. Descriptive statistics were used to summarize the sample and HED prevalence characteristics. Unadjusted and adjusted (including variables significant at p<0.1 in univariate analysis) logistic regression was used to predict HED prevalence. In addition, sex stratified multivariable logistic regression models with HED prevalence were calculated. Based on a previous literature review [5-8,12,16,17,19-31], co-variates included, sex, age group, highest level of education, ethnicity, smoking status, obesity, fruit and vegetable intake, physical activity, sedentary behaviour and health care utilization. Taylor linearization methods were applied to account for the complex study design and the sampling weight. Results from logistic regression analyses are reported as odds ratios (ORs) and 95% confidence intervals (Cis). Missing values (<3.6% on any study variable) were excluded and p<0.05 considered significant.

Round 2

Reviewer 1 Report

Dear Authors,

thank you for the corrections made. Unfortunately, I still feel that the percentages in Table 1 are incorrectly calculated. In my opinion the "missing cases" should be included in the calculations, especially that the number N (%) given in the study and in the table is 4638 or please explain how the % in the "Sample" column was calculated.

Best Regards

Reviewer 2 Report

Even though the authors addressed my comments regarding causal inference based on this cross-sectional analysis, it is not clear how this study contributes to the extensive literature on correlates of heavy drinking, including countries similar to Ecuador. We already know that these factors are correlated with heavy drinking. A causal study with good identification strategy would be publishable.